# Arbitrarily Scalable Environment Generators via Neural Cellular Automata

**Yulun Zhang**[1]  **Matthew C. Fontaine**[2]  **Varun Bhatt**[2]  **Stefanos Nikolaidis**[2]
**Jiaoyang Li**[1]

[1]Robotics Institute, Carnegie Mellon University
[2]Thomas Lord Department of Computer Science, University of Southern California
`yulunzhang@cmu.edu, {mfontain,vsbhatt,nikolaid}@usc.edu,`
`jiaoyangli@cmu.edu`

## Abstract

We study the problem of generating arbitrarily large environments to improve the throughput of multi-robot systems. Prior work proposes Quality Diversity (QD) algorithms as an effective method for optimizing the environments of automated warehouses. However, these approaches optimize only relatively small environments, falling short when it comes to replicating real-world warehouse sizes. The challenge arises from the exponential increase in the search space as the environment size increases. Additionally, the previous methods have only been tested with up to 350 robots in simulations, while practical warehouses could host thousands of robots. In this paper, instead of optimizing environments, we propose to optimize Neural Cellular Automata (NCA) environment generators via QD algorithms. We train a collection of NCA generators with QD algorithms in small environments and then generate arbitrarily large environments from the generators at test time. We show that NCA environment generators maintain consistent, regularized patterns regardless of environment size, significantly enhancing the scalability of multi-robot systems in two different domains with up to 2,350 robots. Additionally, we demonstrate that our method scales a single-agent reinforcement learning policy to arbitrarily large environments with similar patterns. We include the source code at `https://github.com/lunjohnzhang/warehouse_env_gen_nca_public`.

## 1 Introduction

We study the problem of generating arbitrarily large environments to improve the throughput of multi-robot systems. As a motivating example, consider a multi-robot system for automated warehousing where thousands of robots transport inventory pods in a shared warehouse environment. While numerous works have studied the underlying Multi-Agent Path Finding (MAPF) problem [42] to more effectively coordinate the robots to improve the throughput [4, 6, 8, 24, 27, 29, 31, 35, 46], we can also optimize the throughput by designing novel warehouse environments. A well-optimized environment can alleviate traffic congestion and reduce the travel distances for the robots to fulfill their tasks in the warehouse.

A recent work [51] formulates the environment optimization problem as a Quality Diversity (QD) optimization problem and optimizes the environments by searching for the best allocation of shelf and endpoint locations (where endpoints are locations where the robots can interact with the shelves). It uses a QD algorithm to iteratively generate new environments and then repairs them with a Mixed Integer Linear Programming (MILP) solver to enforce domain-specific constraints, such as the storage capacity and connectivity of the environment. The repaired environments are evaluated with an

37th Conference on Neural Information Processing Systems (NeurIPS 2023).

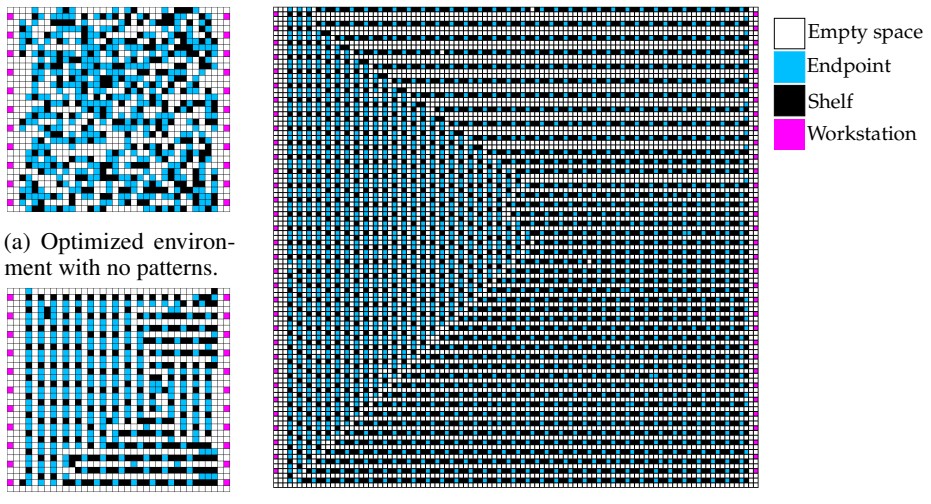

(a) Optimized environment with no patterns.

(b) NCA-generated environment with patterns.

(c) Scaling patterns from the NCA-generated environment to a larger environment.

Figure 1: Examples of optimized warehouse environments. Robots move in between workstations (pink) and endpoints (blue) without traveling through shelves (black) to transport goods. In this environment, the workstations on the left border are used 5 times more frequently than those on the right border. (a) shows a warehouse environment optimized directly with QD algorithms from previous work [51]. It has no obvious regularized patterns. (b) shows an environment generated by our NCA generator with regularized patterns. (c) shows a much larger environment generated by the same NCA generator in (b) with similar patterns.

agent-based simulator [28] that simulates the movements of the robots in the warehouse. Figure 1a shows an example optimized environment, which is shown to have much higher throughput and is more scalable than human-designed environments.

However, with the aforementioned method, the search space of the QD algorithm grows exponentially with the size of the environment. To optimize arbitrarily large environments, QD algorithms require a substantial number of runs in both the agent-based simulator and the MILP solver. Each run also takes more time to finish. For example, in the aforementioned work [51], it took up to 24 hours on a 64-core local machine to optimize a warehouse environment of size only $36 \times 33$ with 200 robots, which is smaller than many warehouses in reality or warehouse simulations used in the literature. For example, Amazon fulfillment centers are reported to have more than 4,000 robots [2]. Previous research motivated by Amazon sortation centers run simulations in environments of size up to $179 \times 69$ with up to 1,000 robots [48, 28].

Therefore, instead of optimizing the environments directly, we propose to train Neural Cellular Automata (NCA) environment generators capable of scaling their generated environments arbitrarily. Cellular automata (CA) [19] are well suited to arbitrary scaling as they incrementally construct environments through local interactions between cells. Each cell observes the state of its neighbors and incorporates that information into changing its own state. NCA [33] represents the state change rules via a neural network that can be trained with any optimization method.

We follow the insight from prior work [10] and use QD algorithms to efficiently train a diverse collection of NCA generators in small environments. We then use the NCA generators to generate arbitrarily large environments with consistent and regularized patterns. In case the generated environments are invalid, we adopt the MILP solver from the previous works [49, 15, 51] to repair the environments. Figure 1b shows an example environment with regularized patterns generated by our NCA generator and then repaired by the MILP solver. Figure 1c shows a much larger environment generated by the same NCA generator with similar patterns and then repaired by MILP. Similar to previous environment optimization methods [51], we need to run the MILP solver repeatedly on small environments (Figure 1b) to train the NCA generator. However, once the NCA generators are trained, we only run the MILP solver once after generating large environments (Figure 1c).

We show that our generated environments have competitive or better throughput and scalability compared to the existing methods [51] in small environments. In large environments where the existing

methods are not applicable due to computational limits, we show that our generated environments have significantly better throughput and scalability than human-designed environments.

We make the following contributions: (1) instead of directly searching for the best environments, we present a method to train a collection of NCA environment generators on small environments using QD algorithms, (2) we then show that the trained NCA generators can generate environments in arbitrary sizes with consistent regularized patterns, which leads to significantly higher throughput than the baseline environments in two multi-robot systems with large sizes, (3) we additionally demonstrate the general-purpose nature of our method by scaling a single-agent RL policy to arbitrarily large environments with similar patterns, maintaining high success rates.

## 2 Background and Related Work

### 2.1 Neural Cellular Automata (NCA)

Cellular automata (CA) [3, 39] originated in the artificial life community as a way to model incremental cell development. A CA consists of grid cells and an update rule for how to change a cell based on the state of its neighbors. Later work [33] showed that these rules could be encoded by a convolutional neural network. By iteratively updating a grid of cells, an NCA is capable of producing complex environments. Figure 6 in Appendix A shows an example NCA environment generation process of 50 iterations from an initial environment shown in Figure 6a.

Representing the rules of CAs by a neural network facilitates learning useful rules for the cellular automata. Several works have trained NCA generators to generate images [33], textures [36, 32], and 3D structures [43]. Other works have replaced the convolutional network with other model architectures [44, 18]. In the case of environment optimization, the objective is computed from a non-differentiable agent-based simulator. Therefore, we choose derivative-free optimization methods to train the NCA generators for environment optimization.

Prior work [10] shows that derivative-free QD algorithms can efficiently train a diverse collection of NCA video game environment generators. However, our work differs in the following ways: (1) we show how environments generated by NCA generators can scale to arbitrary sizes with similar regularized patterns, (2) in addition to encoding the constraints as part of the objective function of QD algorithms, we also use MILP to enforce the constraints on the generated environments, (3) the objective function of the previous work focus on the reliability, diversity, and validity of the generated environments, while we primarily focus on optimizing throughput, which is a simulated agent-based metric of the multi-robot systems.

### 2.2 Quality Diversity (QD) Algorithms

Although derivative-free single-objective optimization algorithms such as Covariance Matrix Adaptation Evolutionary Strategy (CMA-ES) [20] have been used to optimize derivative-free objective functions, we are interested in QD algorithms because they generate a diverse collection of solutions, providing users a wider range of options in terms of the diversity measure functions.

Inspired by evolutionary algorithms with diversity optimization [7, 25, 26], QD algorithms simultaneously optimize an objective function and diversify a set of diversity measure functions to generate a diverse collection of high-quality solutions. QD algorithms have many different variants that incorporate different optimization methods, such as model-based approaches [1, 17, 50], Bayesian optimization [23], and gradient descent [11].

**MAP-Elites.** MAP-Elites [7, 34] constructs a discretized measure space, referred to as *archive*, from the user-defined diversity measure functions. It tries to find the best solution, referred to as *elite*, in each discretized cell. The QD-score measures the quality and diversity of the elites by summing the objective values of elites in the archive. MAP-Elites generates new solutions either randomly or by mutating existing elites. It then evaluates the solutions and replaces existing elites in the corresponding cells with superior ones. After a fixed number of iterations, MAP-Elites returns the archive with elites.

**CMA-MAE.** We select Covariance Matrix Adaptation MAP-Annealing (CMA-MAE) [13, 14] as our QD method for training NCA generators because of its state-of-the-art performance in continuous domains. CMA-MAE extends MAP-Elites by incorporating the self-adaptation mechanisms of

CMA-ES [20]. CMA-ES maintains a Gaussian distribution, samples from it for new solutions, evaluates them, and then updates the distribution towards the high-objective region of the search space. CMA-MAE incorporates this mechanism to optimize the QD-score. In addition, CMA-MAE implements an archive updating mechanism to balance exploitation and exploration of the measure space. The mechanism introduces a threshold value to each cell in the archive, which determines whether a new solution should be added. The threshold values are iteratively updated via an archive learning rate, with lower learning rates focusing more on exploitation. Because of the updating mechanism, some high-quality solutions might be thrown away. Therefore, CMA-MAE maintains two archives, an *optimization archive* implementing the updating mechanism, and a separate *result archive* that does not use the mechanism and stores the actual elites.

## 2.3 Automatic Environment Generation and Optimization

Automatic environment generation has emerged in different research communities for various purposes such as generating diverse game content [41], training more robust RL agents [40, 22], and generating novel testing scenarios [1, 12].

In the MAPF community, Zhang *et al.* [51] have proposed an environment optimization method based on QD algorithms MAP-Elites [7, 34] and Deep Surrogate Assisted Generation of Environments (DSAGE) [1] to improve the throughput of the multi-robot system in automated warehouses. In particular, they represent environments as tiled grids and use QD algorithms to repeatedly propose new warehouse environments by generating random environments or mutating existing environments by randomly replacing the tiles. Then they repair the proposed environments with a MILP solver and evaluate them in an agent-based simulator. The evaluated environments are added to an archive to create a collection of high-throughput environments.

However, since the proposed method directly optimizes warehouse environments, it is difficult to use the method to optimize arbitrarily large environments due to (1) the exponential growth of the search space and (2) the CPU runtime of the MILP solver and the agent-based simulator increase significantly. In comparison, we optimize environment generators instead of directly optimizing environments. In addition, we explicitly take the regularized patterns into consideration which allows us to generate arbitrarily large environments with high throughput.

## 3 Problem Definition

We define the environments and the corresponding environment optimization problem as follows.

**Definition 1** (Environment). *We represent each environment as a 2D four-neighbor grid, where each tile can be one of $N_{type}$ tile types. $N_{type}$ is determined by the domains.*

**Definition 2** (Valid Environment). *An environment is valid iff the assignment of tile types satisfies its domain-specific constraints.*

For example, the warehouse environments shown in Figure 1 have $N_{type} = 4$ (endpoints, workstations, empty spaces, and shelves). One example domain-specific constraint for warehouse environments is that all non-shelf tiles should be connected so that the robots can reach all endpoints.

**Definition 3** (Environment Optimization). *Given an objective function $f : \mathbf{X} \to \mathbb{R}$ and a measure function $\mathbf{m} : \mathbf{X} \to \mathbb{R}^m$, where $\mathbf{X}$ is the space of all possible environments, the environment optimization problem searches for valid environments that maximize the objective function $f$ while diversifying the measure function $\mathbf{m}$.*

## 4 Methods

We extend previous works [10, 51] to use CMA-MAE to search for a diverse collection of NCA generators with the objective and diversity measures computed from an agent-based simulator that runs domain-specific simulations in generated environments. Figure 2 provides an overview of our method. We start by sampling a batch of $b$ parameter vectors $\boldsymbol{\theta}$ from a multivariate Gaussian distribution, which form $b$ NCA generators. Each NCA generator then generates one environment from a fixed initial environment, resulting in $b$ environments. We then repair the environments using a MILP solver to enforce domain-specific constraints. After getting the repaired environments, we

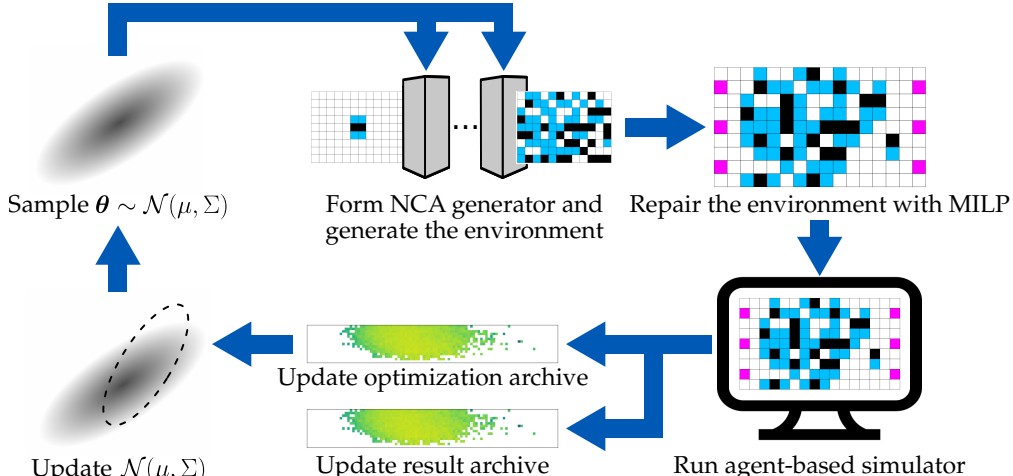

Figure 2: Overview of our method of using CMA-MAE to train diverse NCA generators.

evaluate each of them by running an agent-based simulator for $N_e$ times, each with $T$ timesteps, and compute the average objective and measures. We add the evaluated environments and their corresponding NCA generators to both the optimization archive and the result archive. Finally, we update the parameters of the multivariate Gaussian distribution (i.e., $\mu$ and $\Sigma$) and sample a new batch of parameter vectors, starting a new iteration. We run CMA-MAE iteratively with batch size $b$, until the total number of evaluations reaches $N_{eval}$.

**NCA Generator.** We define an NCA generator as a function $\mathbf{g}(\mathbf{s}; \boldsymbol{\theta}, C) : \mathbf{X} \to \mathbf{X}$, where $\mathbf{X}$ is the space of all possible environments, $\mathbf{s} \in \mathbf{X}$ is a fixed initial environment, and $C \in \mathbb{Z}^+$ is a fixed number of iterations for the NCA generator. $\mathbf{g}$ is parameterized by $\boldsymbol{\theta} \in \boldsymbol{\Theta}$. Since both $\mathbf{s}$ and $C$ are fixed, each parameter vector $\boldsymbol{\theta} \in \boldsymbol{\Theta}$ corresponds to an NCA generator, which corresponds to an environment $\mathbf{x} \in \mathbf{X}$. We discretize the measure space into $M$ cells to create an archive and attempt to find the best NCA generator in each cell to optimize the QD-score of the archive.

Our NCA generator is a convolutional neural network (CNN) with the same architecture as the previous work [10] with 3 convolutional layers of kernel size $3 \times 3$ followed by ReLU or sigmoid activations. Figure 8 in Appendix C.1.1 shows the architecture. We represent the 2D tile-based environments as 3D tensors, with each tile as a one-hot encoded vector. Depending on the tile types, our NCA model has about $1,500 \sim 3,000$ parameters. The input and output of the generator are one-hot environments of the same size, facilitating iterative environment generation by feeding the generator's output back into it. By using a CNN, we can use the same NCA generator to generate environments of arbitrary sizes.

**MILP Repair.** The environments generated by the NCA generators might be invalid. Therefore, we follow previous works [49, 15, 51] to repair the invalid environments using a MILP solver. For each unrepaired environment $\mathbf{x}_{in} = \mathbf{g}(\mathbf{s}; \boldsymbol{\theta}, C)$, we find $\mathbf{x}_{out} \in \mathbf{X}$ such that (1) the hamming distance between $\mathbf{x}_{in}$ and $\mathbf{x}_{out}$ is minimized, and (2) the domain-specific constraints are satisfied. We introduce the domain-specific constraints in detail in Section 5 and Appendix B.

**Objective Functions.** We have two objective functions $f_{opt}$ and $f_{res}$ for the optimization archive and the result archive, respectively. $f_{res}$ runs an agent-based simulator for $N_e$ times and returns the average throughput. $f_{opt}$ optimizes the weighted sum of $f_{res}$ and a similarity score $\Delta$ between the unrepaired and repaired environments $\mathbf{x}_{in}$ and $\mathbf{x}_{out}$. Specifically, $f_{opt} = f_{res} + \alpha \cdot \Delta$, where $\alpha$ is a hyperparameter that controls the weight between $f_{res}$ and $\Delta$. We incorporate the similarity score to $f_{opt}$ so that the NCA generators are more inclined to directly generate valid environments with desired regularized patterns, reducing reliance on the MILP solver for pattern generation. We show in Section 6 that incorporating the similarity score to $f_{opt}$ significantly improves the scalability of the environments generated by the NCA generators.

Let $n$ be the number of tiles in the environment, the similarity score $\Delta$ is computed as:

$$\Delta(\mathbf{x}_{in}, \mathbf{x}_{out}) = \frac{\sum_{i=1}^{n} e_i p_i}{P}, \text{ where } e_i = \begin{cases} 1 & \text{if } (\mathbf{x}_{in})_i = (\mathbf{x}_{out})_i \\ 0 & \text{otherwise.} \end{cases} \text{ and } P = n \cdot \max_i p_i \quad (1)$$

$e_i$ encodes if the tile type of tile $i$ in $\mathbf{x}_{in}$ and $\mathbf{x}_{out}$ are the same. $p_i$ assigns a weight to each tile. As a result, the numerator computes the unnormalized similarity score. $P$ is the theoretical upper bound of the unnormalized similarity score of all environments of a domain. Dividing by $P$ normalizes the score to the range of 0 to 1. Intuitively, the similarity score quantifies the level of similarity between the unrepaired and repaired environments $\mathbf{x}_{in}$ and $\mathbf{x}_{out}$, with each tile weighted according to $p_i$.

**Environment Entropy Measure.** We want the NCA generators to generate environments with regularized patterns. To quantify these patterns, we introduce environment entropy as a diversity measure, inspired by previous work [30, 16]. We define a *tile pattern* as one possible arrangement of a $2 \times 2$ grid in the environment. Then we count the occurrence of all possible tile patterns in the given environment, forming a tile pattern distribution. The *environment entropy* is calculated as the entropy of this distribution. In essence, the environment entropy measures the degree of regularization of patterns within a given environment. A lower value of environment entropy indicates a higher degree of pattern regularization. We use environment entropy as a diversity measure to find NCA generators that can generate a broad spectrum of environments of varying patterns.

## 5  Domains

We perform experiments in three different domains: (1) a multi-agent warehouse domain [28, 51], (2) a multi-agent manufacturing domain, and (3) a single-agent maze domain [16, 5, 1].

**Warehouse.** The warehouse domain simulates automated warehousing scenarios, where robots transport goods between shelves in the middle and human-operated workstations on the left and right borders. We use two variants of the warehouse domain, namely *warehouse (even)* and *warehouse (uneven)*. In warehouse (even), all workstations are visited with equal frequency, while in warehouse (uneven), the workstations on the left border are visited 5 times more frequently than those on the right border. We constrain the warehouse environments such that all non-shelf tiles are connected and the number of shelves is $N_s$ to keep consistent storage capability. We use $f_{opt}$ and $f_{res}$ introduced in Section 4 as the objective functions, and environment entropy as one of the measures. Inspired by previous work [51], we use the number of connected shelf components as the other measure. Figure 1 and Figure 7a in Appendix B show example warehouse environments.

**Manufacturing.** The manufacturing domain simulates automated manufacturing scenarios, where robots go to different kinds of manufacturing workstations in a fixed order to manufacture products. We create this domain to show that our method can generate environments with more tile types for more complex robot tasks. We assume 3 kinds of workstations to which the robots must visit in a fixed order and stay for different time duration. We constrain the manufacturing environments such that all non-workstation tiles are connected and there is at least one workstation of each type. We use $f_{opt}$ and $f_{res}$ introduced in Section 4 as the objective functions, and the environment entropy and the number of workstations as the measures. The number of workstations can approximate the cost of the manufacturing environment. By diversifying the number of workstations, our method generates a collection of environments with different price points. Figure 4 and Figure 7b in Appendix B show example manufacturing environments.

**Maze.** We conduct additional experiments in the maze domain to show that our method can scale a single-agent reinforcement learning (RL) policy trained in small environments to larger environments with similar patterns. The maze domain does not have domain-specific constraints. We use the same objective and measure functions in the previous work [1]. Specifically, $f_{opt}$ and $f_{res}$ are both binary functions that return 1 if the environment is solvable and 0 otherwise. The measures are the number of walls and the average path length of the agent over $N_e$ simulations. Figure 5 and Figure 7c in Appendix B show example maze environments.

We include more detailed information on the domains in Appendix B.

| Domain | $S$ | $S_{eval}$ | $N_s$ | $N_{s\_eval}$ | $N_a$ | $N_{a\_eval}$ | $N_e$ | $b$ | $N_{eval}$ |
|---|---|---|---|---|---|---|---|---|---|
| Warehouse (even) | $36 \times 33$ | $101 \times 102$ | 240 | 2,250 | 200 | 1,400 | 5 | 50 | 10,000 |
| Warehouse (uneven) | $36 \times 33$ | $101 \times 102$ | 240 | 2,250 | 200 | 1,000 | 5 | 50 | 10,000 |
| Manufacturing | $36 \times 33$ | $101 \times 102$ | N/A | N/A | 200 | 1,800 | 5 | 50 | 10,000 |
| Maze | $18 \times 18$ | $66 \times 66$ | N/A | N/A | 1 | 1 | 50 | 150 | 100,000 |

Table 1: Summary of the experiment setup. Columns 2-7 show the configurations related to the environment, and columns 8-10 show the parameters of CMA-MAE. $S$, $N_s$, and $N_a$ are the size of the environments, the number of shelves in the warehouse domain, and the number of agents used in training, respectively. $S_{eval}$, $N_{s\_eval}$, and $N_{a\_eval}$ are their counterparts used in evaluation. We choose $N_{a\_eval}$ to be large enough such that the human-designed environments of the same size are congested while our NCA-generated ones are not. Therefore, the value of $N_{a\_eval}$ depends on the tasks of the robots in each domain.

# 6 Experimental Evaluation

Table 1 summarizes the experiment setup. We train the NCA generators with environments of size $S$ and then evaluate them in sizes of both $S$ and $S_{eval}$. In addition, we set the number of NCA iterations $C = 50$ for environments of size $S$ and $C_{eval} = 200$ for those of size $S_{eval}$ for all domains. $C_{eval}$ is larger than $C$ because the NCA generators need more iterations to converge to a stable state while generating environments of size $S_{eval}$. For the warehouse and manufacturing domains, we use Rolling-Horizon Collision Resolution (RHCR) [28], a state-of-the-art centralized lifelong MAPF planner, in the simulation. We run each simulation for $T = 1,000$ timesteps during training and $T_{eval} = 5,000$ timesteps during evaluation and stop early in case of *congestion*, which happens if more than half of the agents take wait actions at the same timestep. For the maze domain, we use a trained ACCEL agent [9] in the simulations. We include more details of the experiment setup, compute resources, and implementation in Appendix C.

## 6.1 Multi-Agent Domains

**Baseline Environments.** We consider two types of baseline environments for the multi-agent domains (warehouse and manufacturing), namely those designed by human and those optimized by DSAGE [51], the state-of-the-art environment optimization method. In DSAGE, we use $f_{res}$ as the objective function for the result archive and $f_{opt}$ with $\alpha = 5$ for the surrogate archive. With size $S_{eval}$, however, it is impractical to run DSAGE due to the computational limit. One single run of MILP repair, for example, could take 8 hours in an environment of size $S_{eval}$. We use the human-designed warehouse environments from previous work [29, 28, 51]. The manufacturing domain is new, so we create human-designed environments for it.

**NCA-generated Environments.** We take the best NCA generator from the result archive to generate environments of size $S$ and $S_{eval}$. In the manufacturing domain, we additionally take the best NCA generator that generate an environment of size $S$ with a similar number of workstations as the DSAGE-optimized environment. This is to ensure a fair comparison, as we do not constrain the number of workstations, which is correlated with throughput, in the manufacturing domain.

We show more details of the baselines and NCA-generated environments in Appendix D.

### 6.1.1 Environments of Size $S$

Columns 3 and 4 of Table 2 show the numerical results with $N_a = 200$ agents in the warehouse and manufacturing domains with environment size $S$. All NCA-generated environments have significantly better throughput than the baseline environments. In addition, we observe that larger $\alpha$ values have no significant impact on the throughput with $N_a$ agents.

We also analyze the scalability of the environments by running simulations with varying numbers of agents and show the throughput in Figures 3a to 3c. In both the warehouse (even) and manufacturing domains, our NCA-generated environments have higher throughput than the baseline environments with an increasingly larger gap with more agents. In the warehouse (uneven) domain, however, the DSAGE-optimized environment displays slightly better scalability than those generated by NCA. This could be due to a need for more traversable tiles near the popular left-side workstations to alleviate congestion, a pattern creation task that poses a significant challenge for NCA generators

| Domain | Algorithm | Size $S$ with $N_a$ agents | | Size $S_{eval}$ with $N_{a\_eval}$ agents | |
|---|---|---|---|---|---|
| | | Success Rate | Throughput | Success Rate | Throughput |
| warehouse (even) | CMA-MAE + NCA ($\alpha = 0$) | **100%** | **6.79 ± 0.00** | 0% | N/A |
| | CMA-MAE + NCA ($\alpha = 1$) | **100%** | 6.73 ± 0.00 | 0% | N/A |
| | CMA-MAE + NCA ($\alpha = 5$) | **100%** | 6.74 ± 0.00 | **90%** | **16.01 ± 0.00** |
| | DSAGE ($\alpha = 5$) | **100%** | 6.35 ± 0.00 | N/A | N/A |
| | Human | 0% | N/A | 0% | N/A |
| warehouse (uneven) | CMA-MAE + NCA ($\alpha = 0$) | **100%** | **6.89 ± 0.00** | 62% | **12.32 ± 0.00** |
| | CMA-MAE + NCA ($\alpha = 1$) | **100%** | 6.70 ± 0.00 | 8% | 11.56 ± 0.01 |
| | CMA-MAE + NCA ($\alpha = 5$) | **100%** | 6.82 ± 0.00 | **84%** | 12.03 ± 0.00 |
| | DSAGE ($\alpha = 5$) | **100%** | 6.40 ± 0.00 | N/A | N/A |
| | Human | 0% | N/A | 0% | N/A |
| Manufacturing | CMA-MAE + NCA ($\alpha = 5$, opt) | 94% | **6.82 ± 0.00** | **100%** | **23.11 ± 0.01** |
| | CMA-MAE + NCA ($\alpha = 5$, comp DSAGE) | 98% | 6.61 ± 0.00 | N/A | N/A |
| | DSAGE ($\alpha = 5$) | 28% | 5.61 ± 0.12 | N/A | N/A |
| | Human | **100%** | 5.92 ± 0.00 | 0% | N/A |

Table 2: Success rates and throughput of environments of sizes $S$ and $S_{eval}$. We run 50 simulations for all environments except for the manufacturing environments of size $S_{eval}$, for which we run 20 simulations. The success rate is calculated as the percentage of simulations that end without congestion. We measure the throughput of only successful simulations and report both its average and standard error.

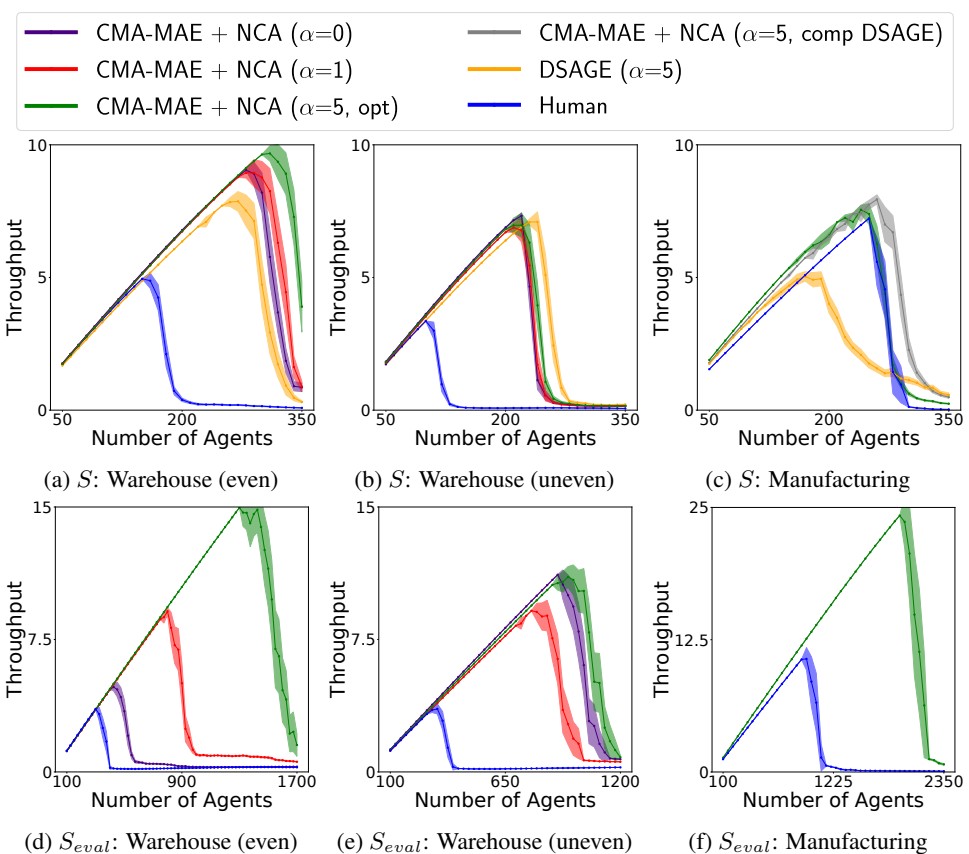

Figure 3: Throughput with an increasing number of agents in environments of size $S$ (a-c) and $S_{eval}$ (d-f). For size $S$, we run 50 simulations and increase the number of agents by a step size of 10. For $S_{eval}$, we run 50 and 20 simulations and increase the number of agents by step sizes of 25 and 50 in warehouse and manufacturing domains, respectively. The solid lines are the average throughput while the shaded area shows the 95% confidence interval.

compared to DSAGE's direct tile search. In addition, in the manufacturing domain, the optimal NCA generator (green line) creates an environment with top throughput for 200 agents, but it is slightly less scalable than the sub-optimal one used for DSAGE comparison (grey line). We conjecture this is due to the optimizer being caught in a local optimum region due to the complexity of the tasks. On the effect of $\alpha$ values, we observe that larger $\alpha$ values enhance the scalability of the generated

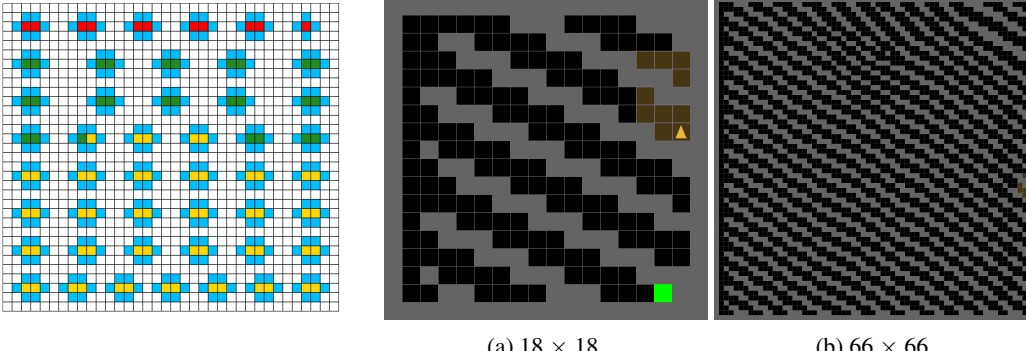

Figure 4: Example manufacturing environment.

Figure 5: Maze environments with similar patterns of different sizes generated by the same NCA generator.

(a) 18 × 18.      (b) 66 × 66.

environments in the warehouse (even) domain, as shown in Figure 3a. We include further discussion of $\alpha$ values in Appendix E.1.

To further understand the gap of throughput among different environments of size $S$ with $N_a$ agents, we present additional results on the number of finished tasks over time in Appendix E.2. All NCA-generated environments maintain a stable number of finished tasks throughout the simulation.

### 6.1.2 Environments of Size $S_{eval}$

Columns 5 and 6 of Table 2 compare environments of size $S_{eval}$ with $N_{a\_eval}$ agents. In the warehouse (even) domain, all environments, except for the one generated by CMA-MAE + NCA ($\alpha = 5$), run into congestion. In the warehouse (uneven) domain, CMA-MAE + NCA ($\alpha = 5$) also achieves the best success rate, even though it does not have the highest throughput. This shows the benefit of using a larger $\alpha$ value with CMA-MAE in creating large and scalable environments.

To further understand the scalability, Figures 3d to 3f show the throughput of the environments with an increasing number of agents. We observe that all our NCA-generated environments are much more scalable than the human-designed ones. In particular, the environments generated by CMA-MAE + NCA ($\alpha = 5$) yield the best scalability in all domains. This is because a higher value of $\alpha$ makes CMA-MAE focus more on minimizing the disparity between the unrepaired and repaired environments. As a result, the optimized NCA generators are more capable of generating environments with desired regularized patterns directly. On the other hand, a smaller value of $\alpha$ makes the algorithm rely more on the MILP solver to generate patterns, introducing more randomness as the MILP solver arbitrarily selects an optimal solution with minimal hamming distance when multiple solutions exist. We show additional results of scaling the same NCA generators to more environment sizes in Appendix E.3. We also include a more competitive baseline by tiling environments of size $S$ to create those of size $S_{eval}$ in Appendix E.4.

### 6.2 Scaling Single-Agent RL Policy

In the maze domain, we show that it is possible to scale a single-agent RL policy to larger environments with similar regularized patterns. We use the NCA generator trained from 18 × 18 maze environments to generate a small 18 × 18 and a large 66 × 66 maze environment, shown in Figure 5. Running the ACCEL [9] agent 100 times in these environments results in success rates of 90% and 93%, respectively. The high success rate comes from the fact that the similar local observation space of the RL policy enables the agent to make the correct local decision and eventually arrive at the goal.

For comparison, we generate 100 baseline 66 × 66 maze environments by restricting the wall counts to be the same and the path lengths to range between 80% to 120% of the one in Figure 5b, to maintain similar difficulty levels. Figure 15 in Appendix D.2 shows two example baseline environments. Testing the ACCEL agent in these environments yields an average success rate of 22% (standard error: 3%), significantly lower than that achieved in the NCA-generated environments. This demonstrates

| | | Size $S$ with $N_a$ agents | | | Size $S_{eval}$ with $N_{a\_eval}$ agents | | |
|---|---|---|---|---|---|---|---|
| Domain | Algorithm | $T_{NCA}$ | $T_{MILP}$ | $\Delta$ | $T_{NCA}$ | $T_{MILP}$ | $\Delta$ |
| | CMA-MAE ($\alpha = 0$) | 0.06 | 2.72 | 0.94 | 0.12 | 228.80 | 0.95 |
| Warehouse (even) | CMA-MAE ($\alpha = 1$) | 0.06 | 2.97 | 0.86 | 0.13 | 9278.18 | 0.84 |
| | CMA-MAE ($\alpha = 5$) | 0.06 | 2.55 | 0.97 | 0.13 | 19.79 | 0.99 |
| | CMA-MAE ($\alpha = 0$) | 0.06 | 2.41 | 0.85 | 0.13 | 1,376.62 | 0.85 |
| Warehouse (uneven) | CMA-MAE ($\alpha = 1$) | 0.06 | 2.36 | 0.95 | 0.13 | 26.54 | 0.96 |
| | CMA-MAE ($\alpha = 5$) | 0.06 | 13.42 | 0.84 | 0.13 | 15,656.04 | 0.86 |
| Manufacturing | CMA-MAE ($\alpha = 5$, opt) | 0.07 | 3.04 | 0.17 | 0.18 | 15,746.10 | 0.18 |

Table 3: CPU runtime measured (in seconds) for the trained NCA generators to generate environments ($T_{NCA}$), and for the MILP solver to subsequently repair the generated environments ($T_{MILP}$) of size $S$ and $S_{eval}$. $\Delta$ refers to the similarity score between the repaired and unrepaired environments. We measure the CPU runtime in machine (1) listed in Appendix C.3 and compute constraints in Appendix C.4.

the ability of our method to generate arbitrarily large environments with similar patterns in which the trained RL agent performs well.

### 6.3 NCA Generation Time

We show the total CPU runtime of generating and repairing the environments of sizes $S$ and $S_{eval}$ in Table 3. In both sizes, larger similarity scores are correlated with shorter MILP runtime. The similarity scores of the manufacturing environments are significantly lower because we use a larger normalization factor $P$ than that in the warehouse domains. Nevertheless, we note that even the longest combined runtime of the NCA generation and MILP repair ($15,746.10 + 0.18 = 15,746.28$ seconds for the manufacturing domain) on the environments of size $S_{eval}$ is significantly shorter than optimizing the environments of that size directly. This reduction in time is attributed to our method, which involves running the MILP repair $N_{eval}$ times with an environment size of $S$ while training the NCA generators, and then using the MILP solver only once with size $S_{eval}$ while generating the environments. Our method stands in contrast to the previous direct environment optimization method [51] which requires running the MILP repair with the size of $S_{eval}$ for $N_{eval}$ times.

### 6.4 On the Benefit of QD Algorithms

QD algorithms can train a diverse collection of NCA generators. In Appendix E.5, we compare the QD-score and archive coverage of CMA-MAE + NCA and DSAGE. We also benchmark CMA-MAE against MAP-Elites [34, 47] to highlight the benefit of using CMA-MAE in training NCA generators. We observe that CMA-MAE + NCA has the best QD-score and archive coverage in all domains.

Furthermore, QD algorithms can train more scalable NCA generators than single-objective optimizers. In Appendix E.6 we compare CMA-MAE with a single-objective optimizer CMA-ES [20]. We observe that CMA-MAE is less prone to falling into local optima and thus better NCA generators.

## 7 Limitations and Future Work

We present a method to train a diverse collection of NCA generators that can generate environments of arbitrary sizes within a range defined by finite compute resources. In small environments of size $S$, our method generates environments that have competitive or better throughput compared to the state-of-the-art environment optimization method [51]. In large environments of size $S_{eval}$, where existing optimization methods are inapplicable, our generated environments achieve significantly higher throughput and better scalability compared to human-designed environments.

Our work is limited in many ways, yielding many directions for future work. First, our method is only arbitrarily scalable under a range determined by finite compute resources. For example, with the compute resource in Appendices C.3 and C.4, the upper bound of the range is about $S_{eval} = 101 \times 102$. Although this upper bound is significantly higher than previous environment optimization methods, future work can seek to further increase it. Second, we focus on generating 4-neighbor grid-based environments with an underlying undirected movement graph. Future work can consider generating similar environments with directed movement graphs or irregular-shaped environments. We additionally discuss social impacts of our work in Appendix F.

## Acknowledgements

This work used Bridge-2 at Pittsburgh Supercomputing Center (PSC) through allocation CIS220115 from the Advanced Cyberinfrastructure Coordination Ecosystem: Services & Support (ACCESS) program, which is supported by National Science Foundation grants #2138259, #2138286, #2138307, #2137603, and #2138296. In addition, this work was supported by the CMU Manufacturing Futures Institute, made possible by the Richard King Mellon Foundation, and was partially supported by the NSF CAREER Award (#2145077).

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
