# OpenReview forum: "Arbitrarily Scalable Environment Generators via Neural Cellular Automata"
_NeurIPS.cc/2023/Conference — NeurIPS 2023 poster_

### Official Review · Reviewer_ShTu · 2023-06-28

**Soundness:** 2 fair
**Presentation:** 3 good
**Contribution:** 2 fair
**Rating:** 4
**Confidence:** 3

**Summary:**

The authors' goal is to optimize large floorplans for warehouses or manufacturing facilities, where robots need to drive during completion of tasks, while avoiding a congestion of the system.
Their innovation is to use the framework of neural cellular automata (NCA) to generate optimized patterns from an initial pattern. Optimized here means maximizing the number of robots in the system.
The training of the NCA's parameters is done using an unsupervised objective, the troughput of the system for the robot drivers, and simoultaneaously optimizing for the diversity of the solution using quality diversity algorithms.
It is shown that the NCA produce regular floor plans for envirnments larger than seen in their training and also perform well under the desired metrics, while being much more efficient to generate than what existing optimizers can achieve.

**Strengths:**

1. Interesting unsupervised objective for training environments of NCA with a combination of different optimization techniques.
2. The authors found a way of optimizing patterns from small environments that seems to scale/generalize to larger environments of a similar structure.
3. I find the evaluation strategy with the RL agent that scales with the NCA interesting and creative. I have not seen jointly scaling agents and environments before and would find this a generally interesting point to investigate independent of this work (although I might not be aware of existing literature on this topic).

**Weaknesses:**

1. An ablation or discussion is missing towards what can break the scalability and generalization ability to new settings: What happens for non-rectangular environments? What if the positioning of the stations is not as in the training set? More diverse test samples would strengthen the argument, that NCAs truly scale well. Also, at which size does the NCA start to fail?
2. I think the setup explored here has big potential for introspecting the solutions. Can one derive the strategy the NCA learned from its produced examples and directly apply it? I.e. find the scalable simple rules that seem to underly the regular patterns generated?
3. Since quality diversity is a big part of the main draft, it would be interesting to see several examples of generated environments in the main draft.
4. To me, the design choices of experimenting with NCA and quality diversity objectives for warehouse plan optimization are not clearly justified (see question section).

**Questions:**

1. Why should a bigger warehouse plan be more difficult to optimize overall? I understand that the complexity of the search space grows, but why is simply tiling good solutions for small environments not mentioned as a (possibly) good baseline? After all, this is how the NCA environments seem to behave (but I might also be missing an obvious problem here).
2. Isn’t a cool thing about your approach that it possibly also scales to arbitrarily shaped environments? I think it would be interesting to see what happens when the constraints of the warehouse are more intricate, e.g. integrating an old warehouse plan with a new one.
3. How ‘broken’ are the NCA’s plans typically? Is there a lot of extra work that is done by the extra solver?
4. Why is it necessary to use quality diversity as an algorithm for warehouse plan optimization? I can imagine how a diversity of environments is interesting in video games, but why should the diversity of the tiling types (which is the entropy measure here) be necessary? I would rather expect a low entropy to be useful, as repeated patterns in the warehouse probably make manufacturing the hall and plans cheaper and less complex.
5. What was the process of manually designing the human-created solutions?
6. Does congestion occur only during testing on the 5,000 time steps? If not, how do you handle it during training?
7. Do you have an idea of how the environment size scale with respect to the maximum number of robots/tasks achievable without congestion?

Minor:
1. L.14 regularized -> regular (?) … if you truly mean regularized, please explain what is being regularized here.
2. L.41 sortation -> sorting (?)
3. L.49 it would be helpful to clarify that your objective is unsupervised and simply comes from the different losees you define.
4. Overall, an (at least informal) definition of the throughput objective missing in the main text.
5. L.168 what kind of object is $\Theta$ and $\mathbf{\theta}$?
6. I think in L.178 it should be “Due to the locality of the NCA operation…” rather than “By using a CNN …” Whatever mechanism you use to implement, the NCA will behave locally as it is its inherent property. The CNN is just an efficient way of implementing it in ML frameworks.

**Limitations:**

Overall, I think the ideas presented in the paper are interesting and a good work of engineering.
However, to make the paper more relevant and useful to a broad audience, it is missing a more abstract perspective on the interesting techniques it proposes.
For this, it should be possible for other researchers to better understand the working principle so that they can estimate how well the techniques could work in different settings. I think this requires more introspection on the results than in the current draft.
Moreover, to make the claim that the NCAs are ‘arbitrarily scalable’, as in the title, the evaluation is lacking different levels of floor plan scales, NCA time scales and an analysis of the regular patterns obtained from the NCA.
For the application to the warehouses, I do not find it very comprehensible that quality diversity plays a big role as emphasized in the paper.
These factors are what lead me to my score.

---

> ### Author Rebuttal · Authors · 2023-08-09
>
> 1. Why should a bigger warehouse plan be more difficult to optimize overall? I understand that the complexity of the search space grows, but why is simply tiling good solutions for small environments not mentioned as a (possibly) good baseline? After all, this is how the NCA environments seem to behave (but I might also be missing an obvious problem here).
>
>     Please see **More Competitive Baselines** in the general rebuttal.
>
> 2. Isn’t a cool thing about your approach that it possibly also scales to arbitrarily shaped environments? I think it would be interesting to see what happens when the constraints of the warehouse are more intricate, e.g. integrating an old warehouse plan with a new one.
>
>     We thank the reviewer for pointing out arbitrarily shaped environments. One naive way of considering non-rectangular shaped environments is encoding the shape of the environments as additional constraints in the MILP solver. We admit in Section 7 that generating environments with irregular shape is one of the future directions of our work.
>
> 3. How ‘broken’ are the NCA’s plans typically? Is there a lot of extra work that is done by the extra solver?
>
> 	Please see **Role of MILP** in the general rebuttal.
>
> 4. Why is it necessary to use quality diversity as an algorithm for warehouse plan optimization? I can imagine how a diversity of environments is interesting in video games, but why should the diversity of the tiling types (which is the entropy measure here) be necessary? I would rather expect a low entropy to be useful, as repeated patterns in the warehouse probably make manufacturing the hall and plans cheaper and less complex.
>
>     We argue that simultaneously optimizing an objective function and diversifying measure functions with a QD algorithm result in better solutions than merely optimizing an objective function. In Appendix D.3, we compare a popular derivative-free single objective optimization algorithm, CMA-ES, with our chosen QD optimizer, CMA-MAE. Figure 16 and Table 4 shows the scalability of the generated environments and numerical results. While CMA-ES generally matches CMA-MAE in throughput and scalability for environments of size S_train, it significantly lags in scalability for size S_eval. In fact, in the warehouse (even) and the manufacture domains, the MILP solver fails to find valid solutions in size of S_eval with the given computational budget specified in Appendix D.4.
>
> 5. What was the process of manually designing the human-created solutions?
>
>     For the warehouse domains, we take the commonly human-designed environments from previous works [28, 29, 47], which 1 x 10 blocks of shelves are repeated placed. We scale this pattern to create human-designed environments of size S_eval.
>
>     For the manufacture domain, we create the human-designed environments of size S_train by taking design insights from the optimal NCA-generated warehouse environment (shown in figure 7f) and maintaining similar number of workstations with the DSAGE optimized environment. With size of S_eval, we take design insights from the NCA generated manufacture environment of the same size (shown in figure 10b) to create the human-designed environment.
>
>     We describe the detailed process of creating human-designed environments in appendix C.4.
>
> 6. Does congestion occur only during testing on the 5,000 time steps? If not, how do you handle it during training?
>
>     During training, we run 5 simulations with 1000 timesteps for each environment. Following previous work (cited in line 482), we stop the simulation early in case of congestion and return the current throughput as the result. This is because (1) we penalize the environments that run into congestion, and (2) we empirically observe that the number of finished tasks per timestep will quickly drop after congestion occurs, resulting in low throughput.
>
>
> 7. Do you have an idea of how the environment size scale with respect to the maximum number of robots/tasks achievable without congestion?
>
> 	Please see **Scalability in More Environment Sizes** in the general rebuttal.

---

> > ### Comment · Reviewer_ShTu · 2023-08-14
> >
> > Thank you for your response, of which I took note.
> > I am more convinced now, that quality diversity is a viable strategy to find optimal environments.
> > However, the clarifications on the scalability have not convinced me that the environments generated by the NCA are "arbitrarily scalable", as stated in the title. The bottleneck of computational time is the MILP solver, which takes for a tile floor of ~100x100 about 8hrs, which is an order of magnitude larger as the time to find the initialization from the NCA.

---

> > > ### Author Response · Authors · 2023-08-19
> > >
> > > Thank you for reading through our rebuttal. For our response to the claim regarding “arbitrarily scalable”, please see our response under official comment in the general rebuttal section. Thank you!

---

### Official Review · Reviewer_v7gi · 2023-07-04

**Soundness:** 3 good
**Presentation:** 3 good
**Contribution:** 3 good
**Rating:** 7
**Confidence:** 4

**Summary:**

In this work, the authors train neural cellular automata with a quality diversity evolutionary algorithm to grow the 2D environment of a multi-agent automated warehouse, manufacturing domain, and single agent maze domain. They show that the approach is able to generate environments that can scale to different sizes, while keeping important regularities.

**Strengths:**

- NCAs can be trained to generate small environments and then scaled to larger environments without further training, thereby saving computational costs
- Compared to previous methods, the Mixed Integer Linear Programming (MILP) has to only be run once to fix the larger environments
- the combination of NCa and MILP is an interesting idea that could be extended to many other domains that NCAs already have been successfully applied to

**Weaknesses:**

- The approach is only shown in similar 2D domains
- The paper could include more environment generation baselines, beyond DSAGE. For example, a comparison to Compositional Pattern Producing Networks (which could also be scaled to larger environments without further training) could elucidate the importance of growth over time
- It would also be useful to include a baseline in which a larger map is created by just concatenating smaller environments and running MILP. Would those perform equally well?

**Questions:**

- "we define a tile pattern as one possible arrangement of a 2- 2 grid in the environment." -> What about the distribution of global patterns? I assume they would not be captured this way?
- One important question is, how easily could this approach be applied to other domains? Does it work particularly well for the domains in this paper because of their types of patterns? Which other domains could it be applied to in the future and what type of patterns would they need to benefit of, in order to allow the scaling to happen? What about environments that need more global pattern than local ones, which the NCA is particularly good at?
- How important is the MILP in the process? At some point in the paper, it says that the process can take 8 hours, which suggests it can also become a bottleneck. It would be good to include the number of errors that MILP fixed (including the number of enironments that were initially not functional) in the result tables for all methods.

**Limitations:**

Yes for the most part. It would be good to talk a bit more about the potential limitation of NCAs to capture more global patterns.

---

> ### Author Rebuttal · Authors · 2023-08-09
>
> 1. "We define a tile pattern as one possible arrangement of a 2- 2 grid in the environment." -> What about the distribution of global patterns? I assume they would not be captured this way?
>
>     We clarify that the environment entropy measure introduced in Section 4 only measures the local pattern. We first define a tile pattern as one possible arrangement of a 2 by 2 grid, then count the occurrences of all possible tile patterns to form a tile distribution. The environment entropy is calculated as the entropy of the tile distribution. High environment entropy represents low regularity in the patterns while low environment entropy represents high regularity. We are more interested in the local patterns instead of global patterns because warehouse and manufacture system designers care more about resolving congestion in the environment with scalable local patterns. If our method is to be applied to other domains in which global patterns are important, we can develop diversity measures that consider global patterns.
>
> 2. One important question is, how easily could this approach be applied to other domains? Does it work particularly well for the domains in this paper because of their types of patterns? Which other domains could it be applied to in the future and what type of patterns would they need to benefit of, in order to allow the scaling to happen? What about environments that need more global pattern than local ones, which the NCA is particularly good at?
>
>     Please see **Generalizing to Other Domains and Real-World Scenarios** in the general rebuttal.
>
> 3. How important is the MILP in the process? At some point in the paper, it says that the process can take 8 hours, which suggests it can also become a bottleneck. It would be good to include the number of errors that MILP fixed (including the number of environments that were initially not functional) in the result tables for all methods.
>
> 	Please see **Role of MILP** in the general rebuttal.
>
> 4. For the question regarding more comptitive baseline methods such as Compositional Pattern Producing Networks, please see **More Competitive Baselines** in the general rebuttal.

---

> > ### Comment · Reviewer_v7gi · 2023-08-15
> >
> > Thank you for clarifying the questions I had. After running the tile-based baseline and addressing my other main concerns, I'm happy to increase my score.
> >
> > However, I don't agree with the comment "Yet, CPPN, mainly adept at generalizing global patterns, isn't well-suited for our use cases, which focus on local patterns to alleviate robot congestion.".
> >
> > CPPNs are not restricted to just creating global patterns and can in fact generate intricate local patterns as well. For example, in Figure 12 in [1], one can clearly observe repeating local patterns with variations, which could be useful for the domains investigated in this paper. Another example of environment generation with CPPNs can be found in [2]. So I believe it would still be an interesting baseline comparison to run in the future.
> >
> > [1] Stanley, Kenneth O. "Compositional pattern producing networks: A novel abstraction of development." Genetic programming and evolvable machines 8 (2007): 131-162.
> > [2] Team, Open Ended Learning, et al. "Open-ended learning leads to generally capable agents." arXiv preprint arXiv:2107.12808 (2021).

---

> > > ### Author Response · Authors · 2023-08-19
> > >
> > > Thank you very much for the pointers and for increasing the score. We will make sure to point out the connection to CPPNs in the revised version.

---

### Official Review · Reviewer_6eLV · 2023-07-08

**Soundness:** 3 good
**Presentation:** 2 fair
**Contribution:** 2 fair
**Rating:** 6
**Confidence:** 3

**Summary:**

This paper proposes a method for generating diverse sets of environments and solving arbitrarily large environments. The main difference of this method is that the individual in QD algorithms serves as an environment generator rather than an environment, making it capable of handling very large-scale environment generation tasks.

**Strengths:**

1/ The proposed method can generate large environments and significantly improve the throughput of multi-robot system.

2/ Using QD algorithms to solve this problems make sense, and the effectiveness are verified.

**Weaknesses:**

1/ The paper is somewhat difficult to read and lacks sufficient background information. To effectively convey the significance of the problem, the authors should provide a more detailed introduction to the research and add more references. This will help readers who are unfamiliar with the content to better understand the paper

2/ The technical contribution of this work is somewhat limited. The proposed method appears to be an ad-hoc combination of existing techniques, and the authors should emphasize their unique technical contributions or demonstrate the method's importance in real-world applications.

3/ To further strengthen the results, it would be beneficial to include more challenging types of environments and competitive baselines.

**Questions:**

1/ Why are there only two lines in Figure 3-f?

2/ Could you show the diverse behaviors of the different generators?

3/ The problem formulation, i.e., geneate a set of environment generators rather than environments, significantly increases the optimization overhead during the training process. How to balance the cimputation cost and the final performance?

4/ How about using QD algorithms to directly generate a set of environments with different scales, and taking the scales as a part of the diversity measure?

**Limitations:**

Yes.

---

> ### Author Rebuttal · Authors · 2023-08-09
>
> 1. Why are there only two lines in Figure 3-f?
>
>     Our method has a hyperparameter alpha that controls the weighting between throughput and similarity score in the objective function. We studied the effect of hyperparameter alpha in Section 6.1 in the warehouse domains. We observe that alpha = 5 is a reasonable setting that can generate environments which outperform human-designed ones. We therefore run experiments in the manufacture domain with alpha = 5. In the final version, we will add the results with more alpha values to validate our hypothesis.
>
> 2. The problem formulation, i.e., generate a set of environment generators rather than environments, significantly increases the optimization overhead during the training process. How to balance the computation cost and the final performance?
>
>     We clarify that generating a set of environment generators rather than environments does not significantly increase the optimization overhead because the majority of the computation resides in agent simulations. For example, in the warehouse (even) domain, the baseline DSAGE algorithm finishes within 24 hours while our method finishes in 28 hours on machine (2) specified in appendix C.5. The extra time is attributed to the MILP solver because, early in the optimization process, the initial NCA generators tend to generate unrepaired environments with simple patterns that do not satisfy most domain-specific constraints.
>
>  3. How about using QD algorithms to directly generate a set of environments with different scales, and taking the scales as a part of the diversity measure?
>
>     We thank the reviewer for the suggestion. By introducing scale as a diversity measure, the QD optimization problem would be much more challenging because we will be simultaneously optimizing NCA generators that generate environments of different scales. In addition, the large computational requirement for the agent-based simulator and MILP solver still remain with this approach. We acknowledge that this is an interesting direction for future work.
>
> 4. For the question regarding more comptitive baseline methods such as Compositional Pattern Producing Networks, please see **More Competitive Baselines** in the general rebuttal.

---

> > ### Comment · Reviewer_6eLV · 2023-08-15
> >
> > Thank you for your response. It addresses several issues that I have. I will increase my score to 6.
> >
> > After reading other Reviewers' comments and responses, the main issues that I have are:
> >
> > 1. Similar to the comments of Reviewer ShTu, I agree that "arbitrarily scalable" is overclaim.
> >
> > 2. The writing should be further strengthened. See Weakness 1.

---

> > > ### Author Response · Authors · 2023-08-19
> > >
> > > Thank you for reading through our rebuttal and for increasing the score. We will improve the writing in the revised version of the paper. For our response to the claim regarding “arbitrarily scalable”, please see our response under official comment in the general rebuttal section. Thank you!

---

### Official Review · Reviewer_fmHH · 2023-07-17

**Soundness:** 4 excellent
**Presentation:** 3 good
**Contribution:** 2 fair
**Rating:** 8
**Confidence:** 4

**Summary:**

The authors have used the Covariance Matrix Adaptation MAP-Annealing algorithm to generate warehouse environments that are efficient for robots to roam around and do their task of moving packages from one place to the other. The authors have presented the idea very nicely in step by step manner, making the concepts clear gradually.

It appears that the work is an improvement over previous works by the same authors. So this cannot be called a novel idea or work. Although, the authors have successfully applied the proposed algorithm to the problem at hand, i.e. organizing a warehouse for swift movement of robots.

**Strengths:**

The results presented by the authors are comprehensive and promising. The appendices cover the areas that are left unhandled in the paper. The authors have provided the code to execute. Although, I could not run it due to the time required for it.

**Weaknesses:**

Including Human design as a metric of comparison is odd, because this is very subjective. An expert human being might be able to design an environment that is far better than a machine. Then there will be a question how have you sampled a human being, have you conducted a survey, or have designed a test that can measure the knowledge of a human being? The questions will keep popping up. I am not saying that you cannot compare with a human being, but the idea is when a scientific study is done including human beings, then the appropriate representatives are picked carefully, which is missing here.

The authors should have mentioned which hardware they used for executing the algorithm. They should have also mentioned how much time it took for them to execute all the scenarios. It appears that the simulation step would have taken quite a large amount of time, then have they considered any parallelism?


Following are some mistakes

	- Figure 1
		○ shows an warehouse environment optimized directly with QD algorithms
		○ [correction]
		○  shows a warehouse environment optimized directly with QD algorithms
	- Line 45
		○ are well suited to arbitrary scaling as they incrementally contruct
		○ [correction]are well suited to arbitrary scaling as they incrementally construct
	- Line 56,57
		○ we only run the MILP solver once after generating the large environments.
		○ [correction]
		○ we only run the MILP solver once after generating large environments.
	- Line 80
		○ However, in the case of the environment optimization
		○ [correction]
		○ However, in the case of environment optimization
	- Line 177
		○ The generator’s input and output are indentical sized one-hot environments
		○ [correction]
		○ The generator’s input and output are identical sized one-hot environments
	- Line 223 and onwards
		○ Repeatedly the work "Manufacture" has been used, instead in most places "Manufacturing" should have been used
	- Line 319
		○ of arbitrarily sizes.
		○ [correction]
of arbitrary sizes.

**Questions:**

What is a measure space that you have mentioned at different places starting from line 115, apparently, you have not defined it. Please provide a reference where this term is defined and an appropriate meaning relevant to your context.

**Limitations:**

Yes, the authors have dedicated a complete section to this. The present work is only limited to the square-shaped cells each having 2 to 4 neighbors. They are aiming to find a strategy for organizing irregularly shaped objects, which would be quite challenging.

---

> ### Author Rebuttal · Authors · 2023-08-09
>
> # Measure Space
> Measure space is part of the QD problem definition and we adopt the definition from the DSAGE paper [47]. Specifically, a measure space is a bounded and discretized space defined by the user-specified diversity measure functions. The goal of the QD algorithm is then to simultaneously optimize an objective function and diversity the measure functions, resulting in an archive of solutions in the measure space. In our context, the measure space of each domain is defined by measure functions specified in Section 4 and 5. We will add the definition to the revised version of the paper.
>
> # Human Subject
> Regarding the human-designed environments, we clarify that many prior works in Multi-Agent Path Finding (MAPF) [27, 29, 1*, 2*, 3*] use similar human-designed environments to benchmark the algorithms. Therefore, we take the human-designed environments used in prior works as reasonable representative of huma-designed environments in our study. We include how the human-designed environments are created in Appendix C.4.
>
> # Hardware
> We thank the reviewer for pointing out the importance of parallization in our experiments. We did parallelize all the simulations in our experiments using compute resources specified in Appendix C.5.
>
> ### References
>
> [1*] Van Nguyen, Philipp Obermeier, Tran Cao Son, Torsten Schaub, and William Yeoh. Generalized target assignment and path finding using answer set programming. In Proceedings of the International Joint Conference on Artificial Intelligence (IJCAI), pages 1216–1223, 2017.
>
> [2*] Hang Ma, Jiaoyang Li, T. K. Satish Kumar, and Sven Koenig. Lifelong multi-agent path finding for online pickup and delivery tasks. In Proceedings of the International Conference on Autonomous Agents and Multiagent Systems (AAMAS), pages 837–845, 2017.
>
> [3*] Zhe Chen, Javier Alonso-Mora, Xiaoshan Bai, Daniel D Harabor, and Peter J Stuckey. Integrated task assignment and path planning for capacitated multi-agent pickup and delivery. IEEE Robotics and Automation Letters, 6(3):5816–5823, 2021.

---

> > ### Comment · Reviewer_fmHH · 2023-08-17
> > **Acknowledgement**
> >
> > Thank you for the clarification.

---

### Official Review · Reviewer_YTho · 2023-07-25

**Soundness:** 2 fair
**Presentation:** 3 good
**Contribution:** 2 fair
**Rating:** 6
**Confidence:** 4

**Summary:**

The research paper addresses the problem of generating large environments to improve the throughput of multi-robot systems. While previous work has proposed Quality Diversity (QD) algorithms for optimizing small-scale environments in automated warehouses, these approaches fall short when replicating real-world warehouse sizes. The challenge arises from the exponential increase in the search space as the environment size grows. Additionally, the previous methods have only been tested with up to 350 robots in simulations, whereas practical warehouses could host thousands of robots.

To overcome these limitations, the authors propose a novel approach using Neural Cellular Automata (NCA) environment generators optimized through QD algorithms. Instead of directly optimizing environments, they train a collection of NCA generators with QD algorithms in small environments and then generate arbitrarily large environments from these generators at test time. The key advantage of using NCA generators is that they maintain consistent, regularized patterns regardless of the environment size, significantly enhancing the scalability of multi-robot systems.

The research is divided into three domains: multi-agent warehouses, multi-agent manufacturing scenarios, and single-agent maze environments. In the warehouse and manufacturing domains, the authors compare their NCA-generated environments with human-designed and state-of-the-art optimized environments, demonstrating higher throughput and better scalability in the NCA-generated environments. In the maze domain, they show that their method can scale a single-agent reinforcement learning (RL) policy to larger environments with similar patterns, outperforming baseline environments.

The paper's strengths lie in its originality, proposing a novel combination of NCA generators and QD algorithms to address the scalability challenge, its quality in presenting a systematic evaluation across multiple domains, clarity in the presentation of algorithms and results, and the significance of its potential impact on various multi-robot systems applications.

**Strengths:**

Originality:

The paper introduces a novel approach to address the problem of generating large environments for multi-robot systems, which is distinct from prior work that focused on optimizing relatively small-scale environments.
The utilization of Neural Cellular Automata (NCA) environment generators, optimized through Quality Diversity (QD) algorithms, is a unique and creative combination of techniques, enabling the generation of large environments with regularized patterns.
The introduction of environment entropy as a diversity measure to quantify regularized patterns is an innovative concept, which contributes to understanding and evaluating the environments' structural characteristics.
Quality:

The research paper thoroughly describes the methodology, algorithms, and experiments, providing comprehensive details for replicability and understanding.
The authors present a systematic evaluation across multiple domains, comparing their NCA-generated environments with human-designed and state-of-the-art optimized environments. The results are statistically analyzed and presented, demonstrating the quality of the proposed approach.
The utilization of CMA-MAE, a state-of-the-art optimization algorithm, adds rigor to the paper's technical foundation.
Clarity:

The paper is well-written and organized, making it easy for readers to follow the methodology and results.
Mathematical formulations and notations are clear and appropriately explained, enhancing the paper's accessibility to researchers in the field.
The use of illustrative figures and tables helps in visualizing the concepts and experimental outcomes.
Significance:

The paper addresses an important problem in the field of multi-robot systems, i.e., scalability and efficient generation of large environments.
The proposed NCA environment generators have the potential to impact various applications, such as automated warehouses and manufacturing systems, by providing optimized and scalable environments.
The demonstrated scalability of single-agent RL policies to larger environments with similar patterns indicates broader implications for other agent-based systems and RL tasks.


**Weaknesses:**

Lack of Novelty in the Proposed Method: The paper claims to propose a new method for optimizing Neural Cellular Automata (NCA) environment generators using Quality Diversity (QD) algorithms. However, the paper does not adequately demonstrate the novelty of the proposed method compared to prior works that have used similar optimization techniques for environment generation. The lack of a comprehensive literature review and clear differentiation from previous approaches weakens the paper's contribution.

Insufficient Evaluation of Scalability: While the paper demonstrates improved scalability of NCA-generated environments compared to human-designed environments, it lacks a thorough analysis of scalability as the environment size increases. The experiments focus on relatively small environments (e.g., 36x33) and then evaluate the same NCA generators on larger environments (e.g., 101x102). A more systematic study with varying environment sizes and intermediate steps would provide a clearer understanding of scalability.

Limited Comparison with Baseline Models: The paper compares the NCA-generated environments with human-designed environments and a state-of-the-art optimization method, DSAGE, but does not include comparisons with other baseline models or state-of-the-art methods in related areas of research. Including more comprehensive comparisons would strengthen the paper's findings.

Lack of Real-world Implementation and Validation: The research focuses on simulated environments and agent-based simulations. While these are suitable for initial validation, the lack of real-world implementation and validation on physical multi-robot systems in actual automated warehouses or manufacturing scenarios reduces the practical significance of the proposed method.

Lack of Discussion on Generalizability: The paper focuses on three specific domains (multi-agent warehouse, multi-agent manufacturing, and single-agent maze), but it lacks a discussion of the generalizability of the proposed method to other domains or applications. Addressing the potential limitations and extensions to different scenarios would enhance the paper's broader impact.

Insufficient Analysis of Hyperparameters: The paper briefly mentions the use of hyperparameters, such as ↵, but does not provide a detailed analysis of how these hyperparameters affect the performance of the proposed method. A thorough sensitivity analysis and tuning of hyperparameters would provide more insights into the model's behavior and stability.

Lack of Open-Source Implementation: While the paper describes the proposed method in detail, it does not provide an open-source implementation or a publicly available codebase. Providing access to the code would allow researchers to reproduce the experiments and build upon the proposed method.

**Questions:**

Clarification on Novelty: Can the authors provide a more detailed explanation of the novel aspects of their proposed method compared to prior works in the field of environment generation and optimization? How does their approach differ from existing approaches in terms of algorithmic design and optimization techniques?

Scalability Analysis: Could the authors elaborate on the reasons behind the observed differences in scalability between the warehouse (even) and warehouse (uneven) domains? What factors contribute to the better scalability in one domain compared to the other? Is it possible to improve scalability in the domain with poorer performance?

Comparison with Baseline Models: While the authors have compared their NCA-generated environments with human-designed environments and DSAGE, could they provide additional comparisons with other baseline models or state-of-the-art methods in related research areas? This would help establish the broader significance of their proposed method.

Real-World Applicability: How feasible is the implementation of the proposed method in actual automated warehouse or manufacturing environments? What are the potential challenges or limitations when applying the method to real-world scenarios? Are there any specific requirements or modifications needed for practical implementation?

Generalization to Other Domains: Could the authors discuss the potential applicability of their proposed method to domains beyond the ones tested in this paper? How well does the approach generalize to diverse multi-agent systems and different task settings? What are the potential obstacles or adaptations required for transferring the method to other scenarios?

Sensitivity to Hyperparameters: How sensitive is the proposed method to the choice of hyperparameters, such as ↵? Can the authors provide insights into the effects of different hyperparameter settings on the quality and scalability of the generated environments? Are there specific guidelines for tuning these hyperparameters?

Open-Source Code Availability: Is the authors' codebase publicly available for reproducibility and further research? If not, would the authors consider releasing the code to the research community? Open-sourcing the implementation would facilitate collaboration and promote further advancements in the field.

Realistic Simulation Assumptions: How well do the agent-based simulators used in the experiments represent real-world multi-agent systems? Are there any limitations or simplifications in the simulators that might affect the validity of the results or the generalizability of the findings?

Impact of Environment Size: In the experiments, the authors evaluate the NCA-generated environments on larger sizes (Seval) compared to the training environments (Strain). Could the authors discuss how increasing environment size affects the performance and efficiency of the NCA generators? Are there any specific challenges or benefits associated with generating larger environments?

Practical Use Cases: Can the authors provide examples of potential real-world applications or use cases where the proposed method could be applied to improve multi-robot systems' performance and scalability? What are the envisioned practical benefits of adopting the proposed approach in such scenarios?

**Limitations:**

Simplified Simulations: The agent-based simulators used in the experiments might oversimplify real-world scenarios, leading to potential discrepancies between simulated and actual multi-robot system behaviors. The authors should discuss the implications of these simplifications and any potential deviations from real-world performance.

Lack of Real-World Validation: The proposed method's evaluation is primarily based on simulations, and there is no direct validation in real-world warehouse or manufacturing environments. It would be beneficial to include experiments in real-world scenarios to assess the transferability and effectiveness of the NCA-generated environments.

Generalization to Other Environments: While the paper shows promising results in the warehouse and manufacturing domains, it remains unclear how well the proposed method generalizes to more complex and diverse environments, such as outdoor spaces, multi-floor warehouses, or environments with dynamic obstacles.

Hyperparameter Sensitivity: The method's performance might be sensitive to the choice of hyperparameters, such as the weight ↵ used to balance similarity and objective functions. The authors should provide a more comprehensive analysis of the sensitivity of the method to different hyperparameter settings.

Scalability in All Domains: While the paper demonstrates superior scalability in some domains, the method's performance in other domains, such as the warehouse (uneven) scenario, does not show the same level of improvement over baseline environments. The authors should address the limitations that might hinder scalability in certain domains and propose potential solutions.

Ethical and Societal Considerations: Although the paper focuses on technical aspects, it is essential to discuss the broader societal impacts and potential ethical considerations of deploying large-scale multi-robot systems in various real-world applications. Addressing ethical implications and potential negative societal consequences can provide a more well-rounded perspective on the research.

Reproducibility: While the paper outlines the method's details, there is no mention of whether the authors' codebase is available for replication and further research. Providing open-source code or detailed instructions for replication would improve the study's transparency and reproducibility.

Comparison with More Baseline Models: The paper compares the NCA-generated environments with human-designed environments and DSAGE. Including additional comparisons with other state-of-the-art environment generation and optimization methods would strengthen the research's credibility and position it in the broader context of related work.

---

> ### Author Rebuttal · Authors · 2023-08-09
>
> 1. Clarification on Novelty:
>
>     In prior environment generation methods [47], the size of the environments searched over and the size of the environments generated are the same. These methods involve thousands of agent simulations during the search. In large environments that we generate in our work, each simulation may take up to a day, making it intractable to directly apply prior methods. Hence, searching for environment generators in smaller environments and leveraging them to generate larger environments is a crucial step in scaling environment generation.
>
>     Another work [11] has combined QD algorithm and NCA to generate diverse game environments in small sizes (16 by 16). Our work is different from the following aspects: (1) while the prior work focuses on generating diverse game levels, we show that the NCA generators can be trained at a small scale and then be used to generate larger environments at various scales while maintaining consistent local patterns, which is beneficial to large multi-agent system design, (2) we incorporated the MILP solver to enforce domain specific constraints, (3) our objective function includes the throughput, which is a simulated agent-based metric, while the objective function of the previous work focuses on properties of the environments, (4) we introduced the similarity score S (introduced in section 4) to the objective function so that the generated environments can more easily scale to larger sizes.
>
> 2. Scalability Analysis:
>
>     Environments of warehouse (uneven) domains are generally less scalable because the tasks of the robots are unevenly distributed. In particular, robots are 5 times more likely to go to workstations on the left border than those on the right. As a result, the robots can more easily get congested because more robots will be traveling to the left-border workstations throughout the simulation.
>
> 3. Comparison with Baseline Models:
>
>     Please see **More Competitive Baselines** in the general rebuttal.
>
> 4. Real-World Applicability:
>
>     Please see **Generalizing to Other Domains and Real-World Scenarios** in the general rebuttal.
>
> 5. Generalization to Other Domains:
>
>     Please see **Generalizing to Other Domains and Real-World Scenarios** in the general rebuttal.
>
> 6. Sensitivity to Hyperparameters:
>
>     The most important hyperparameter in our method is alpha, which controls the weighting between throughput and similarity score in the objective function. We show the experimental results of different alpha values in warehouse (even) and warehouse (uneven) domains in section 6.1. In Figure 7 and 9 of Appendix C.3, we show the generated environments trained with different alpha values. In general, larger alpha results in better scalability in larger environments.
>
> 7. Open-Source Code Availability:
>
>     We have included the code with instructions to reproduce our results in the supplemental material. We will make the code public if our paper is accepted.
>
> 8. Realistic Simulation Assumptions:
>
>     Please see **Generalizing to Other Domains and Real-World Scenarios** in the general rebuttal.
>
> 9. Impact of Environment Size:
>
>     The performance of the NCA generator in terms of runtime is neglectable with increasing environment sizes. We provide the detailed runtime of the NCA generator and MILP solver in sizes of S_train and S_eval in table 5 of appendix D.4.
>
> 10. Practical Use Cases:
>
>     Please see **Generalizing to Other Domains and Real-World Scenarios** in the general rebuttal.

---

### Author Rebuttal · Authors · 2023-08-09

We thank all reviewers for the detailed feedback. We appreciate that the reviewers find the combination of NCA, QD, and MILP novel and interesting (Reviewer YTho, v7gi) and the results comprehensive and promising (Reviewer fmHH).

## Role of MILP
Q3 of reviewer v7gi and Q3 of ShTu

The role of MILP varies by the domains. To clarify it, we show the unrepaired and repaired environments of size S_train in Figure 1 in the rebuttal as well as Figure 8e and 8f in Appendix C.3. In the warehouse (even) domain, its role is minimal, merely enforcing constraints due to slight disparities between unrepaired and repaired environments. Conversely, in the warehouse (uneven) and manufacture domains, MILP plays a role in pattern creation in addition to constraint enforcement. For example, shown in Figure 1c, the NCA generator for the warehouse (uneven) domain generates more endpoints (blue) on the left and more storage shelves (black) on the right. This facilitates the MILP in repairing the environment such that the left part has less obstacles (storage shelves), enabling agents to more efficiently access the frequently visited left-border workstations.

For S_train environments, MILP takes up to 1 minute, while for larger S_eval sizes, it's up to 8 hours. This longer duration is a one-time event after training NCA generators. Our method reduces the overhead of frequent large environment repairs when optimizing with size S_eval.

## Scalability in More Environment Sizes
Q7 of reviewer ShTu

To further demonstrate our method’s scalability, we use the trained NCA generators from Section 6 to generate progressively larger environments and run simulations. Figure 2 in the rebuttal shows the result. The y-axis illustrates two metrics: maximum mean throughput over 50 simulations (right) and the maximum scalability, defined as the agent count at this maximum (left).

We see an increasing trend for both maximum scalability and maximum mean throughput as the environment size increases since more space will be available for the agents to move. Layouts generated by our algorithm generally scale better than the human-designed warehouse layouts.

We see two exceptions: Maximum mean throughput in the 69x69 warehouse (uneven) environment and both metrics in the 57x58 manufacture environment. We can attribute this to the interaction between the MILP and the specific environment generated by the NCA. Since MILP makes numerous changes to the generated environment in these domains (e.g. Fig. 1c and 1d in the rebuttal), it is possible that certain combinations of generated environments and MILP random seeds can lead to repaired layouts that create congestion. However, if we encounter such issues in practice, it is possible to either leverage a different NCA generator from the archive or re-run the MILP repair with a different random seed.

## More Competitive Baselines
Q3 of reviewer YTho, Q1 of ShTu, and concerns regarding baseline from reviewers 6eLV and v7gi

Multiple reviewers (YTho, 6eLV, v7gi) suggested adding more baseline methods. We chose two representative baselines in our work: DSAGE [47] (a state-of-the-art technique for multi-agent environment optimization) and human-designed environments. Previous environment generation techniques often have the same searched and generated environment sizes, likely facing computational challenges similar to DSAGE. We thank reviewer v7gi for pointing out the Compositional Pattern Producing Network (CPPN) as another baseline. Yet, CPPN, mainly adept at generalizing global patterns, isn't well-suited for our use cases, which focus on local patterns to alleviate robot congestion.

### Tiling Environment Baseline
Suggested by reviewer v7gi and ShTu, we add a new baseline. We tile the environments of S_train shown in Appendix C.3, Figure 7f (warehouse (even)), 7i (warehouse (uneven), and 8d (manufacture), to create the large S_eval environments. We then use MILP to enforce constraints.

We run 50 simulations with N_a_eval agents specified in table 1 of the paper. The new baseline failed in the warehouse (even) domain and achieved only a 23% success rate in the warehouse (uneven) domain. Figure 3 in the rebuttal displays the S_eval-sized tiled environments and tile-usage maps, which show the frequency of each tile used in the simulation. As shown in Figure 3b and 3d, the agents are congested, resulting in low success rates. In contrast, for the manufacture domain, the baseline matches our method, with a 100% success rate and 22.73 average throughput. This success is attributed to Figure 8d's tiling resembling NCA-generated patterns in Figure 10b. Thus, the tiling baseline may be a good method for the manufacture domain, yet it falls short in the warehouse domains.

## Generalizing to Other Domains and Real-World Scenarios
Q4, 5, 8, 10 of reviewer YTho, and Q2 of v7gi

Large companies such as Amazon and Alibaba have deployed multi-robot systems in warehouses to transport packages or inventory pods. Therefore, one real world application of our method is optimizing the layout of the automated warehouses to improve throughput. Since our method is agnostic to the specific agent simulator and only requires metrics such as throughput post-simulation, we can plug-in different simulators and apply our environment generation algorithm. We selected our simulator because similar ones are used in the prior works [28, 47].

Nevertheless, the potential obstacles of applying our method to the real-world physical scenarios are sim-to-real gaps and availability of a realistic and efficient simulator. Similar to prior works [27, 28, 29, 47], we make simplifications in our agent-based simulators such as perfect robot motion dynamics and deterministic environment dynamics. We are excited about integrating our idea of scalable environment generation via NCAs with physical robots and warehouse simulators in the future.

We will include all the above discussions in the appendix of the revised paper.

---

> ### Author Response · Authors · 2023-08-19
> **Revised Claim of Arbitrary Scaling**
>
> Reviewers ShTu and 6eLV raise an important point that the claim that our method is arbitrarily scalable is an overclaim. After some discussion amongst the authors, we believe the source of the overclaim is claiming we’ve shown that the generators scale to arbitrarily large environments, which we agree is inaccurate.
>
> A more accurate claim is that under finite memory and compute resources, there are a range of scales that our method can generate: from $33 \times 36$ to $101 \times 102$. By selecting an arbitrary layout size within that range, we’ve shown empirically that the generated environments maintain the same regularized patterns at different scales and therefore induce the similar performance across scales (see Figure 2 in the rebuttal pdf). The adjusted claim is our method is arbitrarily scalable within a finite range, rather than able to scale to arbitrarily large environments. However, it is important to note that the upper bound on this range is due to computational budgets. For example, given 8 hours of compute, the upper bound is around $101 \times 102$. Given more compute, our method should scale to larger environments, given the behavior we observe for a fixed range of scales.
>
> In addition, we reiterate that we have greatly reduce the computational overhead of the MILP solver on the S_eval-sized environments. Specifically, with previous environment optimization methods, such as DSAGE, we need to run the MILP solver 10,000 times to optimize the environment of the desired size, each being as long as 8 hours. With our methods, however, we only need to run the MILP solver once after generating the environment. We also agree that reducing the runtime of the MILP solver or studying other more efficient ways of enforcing domain-specific constraints are interesting directions of future works.
>
> We thank reviewers ShTu and 6eLV for raising this point. We will update the claim in the main paper to accurately reflect the scalability claim and add an appendix section discussing the upper bound given finite compute resources. Please let us know if this adjusted claim is satisfactory to all reviewers.

---

### Comment · Area_Chair_ZSKp · 2023-08-15
**Please read and respond to authors' rebuttals**

Dear reviewers,

Thank you for your reviews. The authors have posted their rebuttal. If you have not yet done so, please read the rebuttal and the other reviews, and comment on whether the rebuttal has addressed your comments or concerns.

---

### Decision · Program_Chairs · 2023-09-21

**Decision:**

Accept (poster)

**Comment:**

This paper proposes to optimize Neural Cellular Automata (NCA) environment generators via QD algorithms to generate arbitrarily large environments to improve the throughput of multi-robot systems. All the reviewers agreed that the paper aimed to solve an important problem, and the solution is interesting and original. The reviewers also raised concerns about the quality of writing, overclaims of scalability, simple evaluations, and choice of baselines. The authors' rebuttal has sufficiently addressed most of these concerns. During discussion, most of the reviewers voted for acceptance. Please incorporate your response in rebuttals and reviewer's suggestions, especially around "scalability" into the final version of this paper.